# Thermal Behavior of Mixed Plastics at Different Heating Rates: I. Pyrolysis Kinetics

**DOI:** 10.3390/polym13193413

**Published:** 2021-10-05

**Authors:** Ibrahim Dubdub, Mohammed Al-Yaari

**Affiliations:** Chemical Engineering Department, King Faisal University, P.O. Box 380, Al-Ahsa 31982, Saudi Arabia; idubdub@kfu.edu.sa

**Keywords:** pyrolysis, plastic waste, activation energy, thermogravimetric analyzer (TGA), kinetics, reaction mechanism

## Abstract

The amount of generated plastic waste has increased dramatically, up to 20 times, over the past 70 years. More than 50% of municipal plastic waste is composed of polystyrene (PS), polypropylene (PP), and low-density polyethylene (LDPE) products. Therefore, this work has developed a kinetic model that can fully describe the thermal decomposition of plastic mixtures, contributing significantly towards the efficiency of plastic waste management and helping to save the environment. In this work, the pyrolysis of different plastic mixtures, consisting of PP, PS, and LDPE, was performed using a thermogravimetric analyzer (TGA) at three different heating rates (5, 20, and 40 K/min). Four isoconversional models, namely Friedman, Flynn–Wall–Qzawa (FWO), Kissinger–Akahira–Sunose (KAS), and Starink, have been used to obtain the kinetic parameters of the pyrolysis of different plastic mixtures with different compositions. For the equi-mass binary mixtures of PP and PS, the average values of the activation energies were 181, 144 ± 2 kJ/mol obtained using the Freidman and integral (FWO, KAS, and Starink) models, respectively. However, higher values were obtained for the equi-mass ternary plastic mixtures of PP, PS, and LDPE (Freidman: 255 kJ/mol, FWO: 222 kJ/mol, KAS: 223 kJ/mol, and Starink: 222 kJ/mol). The most suitable reaction mechanisms were obtained using the Coats–Redfern model. The results confirm that the most controlling reaction mechanisms obey the first-order (F1) and the third-order (F3) reactions for the pyrolysis of the equi-mass binary (PS and PP) and equi-mass ternary (PS, PP, and LDPE) mixtures, respectively. Finally, the values of the pre-exponential factor (A) were obtained using the four isoconversional models and the linear relationship between *ln* A and the activation energy was confirmed.

## 1. Introduction

Plastic waste management must be considered as a key priority. The amount of the generated plastic waste has increased in the past 70 years by more than 20 times. In addition, the annual production rate of plastic is much higher than that of plastic recycling; thus, most plastic waste is either disposed of in landfills or incinerated. [1] Moreover, the recycling process has some limitations due to the availability of some additives used to improve the properties of plastic and to meet application needs [2]. Furthermore, plastic production consumes almost 4% of the global oil production rate [3]. Thus, plastic waste is one of the main sources of contamination, with serious consequences for environmental sustainability.

Due to its moderate operating temperatures and clean products, pyrolysis is a preferable option to recover energy from municipal plastic waste (MPW) as chemicals and fuels [4,5,6,7,8].

MPW mainly contains low-density polyethylene (LDPE), high-density polyethylene (HDPE), polypropylene (PP), polystyrene (PS), and polyethylene terephthalate (PET) but with different compositions. Therefore, extensive work has been performed to obtain the kinetic parameters of pure and mixed plastics. 

Wu et al. (1993) [9] studied the pyrolysis of six polymers HDPE, LDPE, PP, PS, polyvinylchloride (PVC), and acrylonitrile butadiene styrene (ABS) of MPS with their mixtures using a thermogravimetric analyzer (TGA) at heating rates of 1, 2, and 5.5 K/min. Insignificant interaction between these polymers was reported during the pyrolysis process. 

Chowlu et al. (2009) [10] studied the pyrolysis behavior of a mixture of PP and LDPE at five different mixture compositions and heating rates. The Vyazovkin (VYA) model, as one of the model-free techniques, was used to investigate the effect of conversion on activation energy. Three different zones were reported: slow at low conversion range, slightly high at the middle range of conversion, and strongly high at high conversion range. Therefore, the best mixture weight ratio of PP/LDPE, with the lowest activation energy, was reported as 65/35.

Aboulkas et al. (2010) [11] studied the reaction mechanism of the thermal decomposition of HDPE, LDPE and PP using the Coats–Redfern and Criado methods. While the contracting sphere model best fit the HDPE and LDPE data, the contracting cylinder model worked well with the PP data. 

Diaz Silvarrey and Phan (2016) [1] investigated the reaction mechanism of the thermal decomposition of five different polymers: HDPE, LDPE, PP, PS, and PET using TGA and MATLAB. Kissinger–Akahira–Sunose (KAS), Malek, and linear model fitting methods were used to obtain the mechanism of the pyrolysis, which was checked using the experimental data obtained from TGA tests. Four heating rates (5, 10, 20, and 40 K/min), covering the temperature range of 30–700 °C, were employed. All the polymer samples were reported to have one step of decomposition, moving to a higher temperature with the order: PS ˂ PET ˂ PP ˂ LDPE ˂ HDPE. Using the KAS method, the following values of activation energy and pre-exponential factor were reported: (PS: 192.61 KJ/mol, 5.52E14 1/K), (HDPE: 202.40 KJ/mol, 3.23E16 1/K), (LDPE: 267.61 KJ/mol, 7.86E19 1/K), and (PP: 261.22 KJ/mol, 3.03E21 1/K). 

Yu et al. (2016) [12] conducted a helpful review on the thermal decomposition of PVC mixed with different polymers (PP, polyethylene (PE), and PS). The interaction between polymers was found to be mainly dependent on the nature of the mixed polymers. 

Anene et al. (2018) [13] studied the thermal degradation of different compositions of LDPE/PP mixtures. The degradation started at a lower temperature for the LDPE/PP mixture than the pure LDPE, proving an interaction between the polymers. 

Mumbach et al. (2019) [14] studied the thermal decomposition of MPW by a TGA under inert conditions at four heating rates (5, 10, 20, and 30 K/min). The feedstock of the MPW included 51.85% PP, 17.28% LDPE, 7.41% HDPE, 17.28% PVC, PET, and PS, and 6.18% lignocellulosic. While the kinetic parameters, such as activation energy, were estimated by four isoconversional (FWO, KAS, Starink, and VYA) models, the reaction mechanisms were obtained by the Criado master plots. The following three main reaction stages were identified: the decomposition of holocellulose with the first stage of the decomposition of PVC (dichlorination); the decomposition of PS and some adhesive acrylic-based resins; the thermal decomposition of PP, LDPE, and HDPE, and the second stage of the decomposition of PVC. 

Recently, Dubdub and Al-Yaari (2020) [15] investigated the co-pyrolysis process of mixed polymers (PS/PP/HDPE/LDPE) at a single heating rate (60 K/min). The Coats–Redfern and Criado models were used to obtain the kinetic parameters and the most suitable reaction mechanism. In addition, a synergetic effect was observed for some mixtures and compositions.

Although different kinetic investigations have been performed for pure and mixed plastics, most are inaccurate and inconsistent because of their simple assumed mechanisms or experiment conditions (single heating rate). Therefore, this investigation has developed a kinetic model that fully describes the thermal behavior of the pyrolysis of mixed plastics, comprising PP, PS, and LDPE, at different heating rates. In addition, the triple kinetic parameters (activation energy, pre-exponential factor, and most suitable controlling mechanism/s) have been determined.

## 2. Materials and Methods

### 2.1. Materials

Pellets of PP, PS, and LDPE, supplied by Ipoh SY Recycle Plastic Sdn. Bhd., Perak, Malaysia, were ground into powder. Then, 10 mg of each powder sample was used throughout the study. Proximate and ultimate analysis was performed to characterize the polymer samples; these data are presented in Table 1. Details of both tests are described elsewhere [15].

### 2.2. Thermal Decomposition Experiments

Pyrolysis of different mixtures of PP, PS, and LDPE with different compositions at three different heating rates (5, 20, and 40 K/min) were performed using the thermogravimetric analyzer (TGA-7), manufactured by PerkinElmer, Shelton, CT, USA, and equipped with a high precision weighing balance. Thermal decomposition experiments were conducted under N_2_ (99.999%) gas flowing at 100 cm^3^/min. The experimental matrix is presented in Table 2.

### 2.3. Kinetic Theory

The reaction rate ($\frac{d\alpha }{dt}$) of the pyrolysis of PVC can be expressed as follows:(1)$$\frac{d\alpha }{dt}=K\left(T\right)\times f\left(\alpha \right)$$
(2)$$\alpha =\frac{w_{o}-w}{w_{o}-w_{f}}$$
where: 

*α*: is the reaction conversion;

*t*: is time (min);

*K*: is the reaction rate constant (K^−1^), expressed as: $K\left(T\right)=A\exp\left(-\frac{E_{a}}{RT}\right)$;

*A*: is a pre-exponential factor (K^−1^); 

*E_a_*: is the activation energy (kJ/mol); 

*R*: is the universal gas constant (8.314 J/mol.K);

*T*: is temperature (K);

*w_o_*: is the initial weight of the sample used for the experiment;

*w*: is the instantaneous weight of the sample (at time t); 

*w_f_*: *is* the weight left of the sample at the end of the experiment.

For non-isothermal pyrolysis, the heating rate (β) can be defined as $\beta =\frac{dT}{dt}$, and thus the reaction rate can be written as:(3)$$\beta \frac{d\alpha }{dT}=A\exp\left(-\frac{E_{a}}{RT}\;\right)\times f\left(\alpha \right)$$

The development of a high-efficient kinetic model that can describe the pyrolysis process requires obtaining kinetic parameters accurately. Using TGA data at different heating rates, the activation energy can be obtained using isoconversional (model-free) models such as the Freidman (Equation (4)), FWO (Equation (5)), KAS (Equation (6)), and Starink (Equation (7)) models. These four models are among the most widely used models and thus have been used in this investigation.
(4)$$\ln\left(\beta \frac{d\alpha }{dT}\right)=\ln\left[A\;f\left(\alpha \right)\right]-\frac{E_{a}}{R\;T}$$
(5)$$\ln\left(\beta \right)=ln\frac{A\;E_{a}}{R\;g\left(\alpha \right)}-5.331-1.052\frac{E_{a}}{R\;T}$$
(6)$$\ln\left(\frac{\beta }{T^{2}}\right)=ln\frac{A\;R}{E_{a}\;g\left(\alpha \right)}-\frac{E_{a}}{R\;T}$$
(7)$$\ln\left(\frac{\beta }{T^{1.92}}\right)=Constant-1.0008\frac{E_{a}}{R\;T}$$

However, model-fitting methods, such as the Coats–Redfern model (Equation (8)), can be used to obtain kinetic parameters based on a hypothetical reaction model.
(8)$$ln\left[\frac{g\left(\alpha \right)}{T^{2}}\right]=ln\left[\frac{A_{\;}R}{\beta \;E}\right]-\frac{E_{a}}{R\;T}$$
where *f(α)* and *g(α)* are the differential and integral forms of the conversion-dependent term, respectively. Table 3 shows different commonly used solid-state thermal reaction mechanisms along with the *f(α)* and *g(α)* expressions. 

In this work, the activation energy values were obtained using four isoconversional models (Equations (4)–(7)) and the TGA experimental data. These models are independent of the reaction mechanism, and they are among the most used models. The most suitable reaction mechanism was determined by the Coats–Redfern model (Equation (8)). After this, the values of the pre-exponential factor were calculated using the isoconversional models’ equations. Finally, the linear relationship between *ln(A)* and *E_a_* was checked. 

## 3. Results and Discussion

### 3.1. Thermogravimetric Analysis 

The thermogravimetric (TG) and derivative–thermogravimetric (DTG) curves of the pyrolysis of mixed polymer samples are presented in the figures below. Although Figure 1 represents data of the equi-mass binary mixture of PP and PS, Figure 2 shows the TGA data of the equi-mass ternary mixture of PP, PS, and LDPE at different heating rates. Generally, as reported for individual plastics in previous works [15,16,17], thermograms of all mixtures have a similar inverted S-shape. However, they were shifted to higher temperatures as the heating rate increased (i.e., a faster heating rate implies a small weight loss at specific temperatures). In addition, as the heating rate increases, the change in the rate of weight loss increases, thus producing higher DTG values. Furthermore, complete pyrolysis (100% weight loss) of all tests has been observed, which reflects the purity of the polymer samples when ash content is negligible (see Table 1). Table 4 presents the characteristic decomposition temperatures (i.e., the onset, peak, and final decomposition temperatures). 

These figures clearly show that there was only one main reaction region for the pyrolysis of mixed polymers. These findings are in full agreement with different published data [12,13].

### 3.2. Activation Energy Determination

As mentioned earlier, four isoconversional models were used to obtain the activation energy, which is the main kinetic parameter by which the optimum mixture composition can be recommended. The linear regression lines of binary equi-mass PP and PS mixtures (tests 1–3) at different conversions ranging from 0.1 to 0.9, using the Freidman, FWO, KAS, and Starink models, are shown in Figure 3. Figure 4 illustrates the lines of ternary equi-mass in PP, PS, and LDPE mixtures (tests 4–6) at the same range of conversion. In addition, the obtained values of activation energy by each model at different conversions for binary and ternary equi-mass mixtures are presented in Table 5 and Table 6, respectively.

Generally, the obtained values of activation energy, from non-isothermal TGA data, by model-free methods are more reliable than those obtained by model-fitting models since the model-free values are mechanism-independent. Activation energy values are obtained as a function of reaction conversion, which may help to show multi-reaction complexity [18]. As the conversion increases, the value of the activation energy also increases [19] (see Figure 5); in this case, the average values can be utilized when industrial processes are modelled. 

Although all models performed well to obtain activation energy (check the regression coefficient values (R^2^) presented in Table 5 and Table 6), the Freidman model provided a slightly different value to the other models. Integral models (FWO, KAS, and Starink) have allow approximations in their mathematical formulations [11]. 

For the equi-mass binary mixtures (tests 1–3), the average values of the activation energies were 181, 144 ± 2 kJ/mol obtained using the Freidman and integral models, respectively. However, higher values were obtained for the equi-mass ternary plastic mixtures of PP, PS, and LDPE (tests 4–6) (Freidman: 255 kJ/mol, FWO: 222 kJ/mol, KAS: 223 kJ/mol, and Starink: 222 kJ/mol). This can be attributed to LDPE activation energy, which is higher than that of pure PP, PS, and the interaction between the mixture components. In addition, at low conversions (α ˂ 0.3), the values of activation energy were almost unstable, caused by errors in the baseline determination [20], or an undetected reaction that had occurred at this low range of temperatures. 

### 3.3. Reaction Mechanism Determination

The Coats–Redfern model has been used to obtain the most appropriate reaction mechanism. By plotting $ln\left[\frac{g\left(\alpha \right)}{T^{2}}\right]$ vs. 1/T for 15 different solid-state reaction mechanisms ($g\left(\alpha \right))$ presented in Table 3, activation energy values at different heating rates were obtained (see Table 7 and Table 8). Results showed excellent linear regression (R^2^ > 0.98). The average values of the activation energy, obtained using the Coats–Redfern model for different reaction mechanisms, were then compared with the average values obtained by the isoconversional models. 

As shown in Table 7, the average value of the activation energy obtained by Coats–Redfern for the first-order reaction mechanism (F1) was 152 kJ/mole, which is the closest value to the average value obtained by the isoconversional models. Thus, the pyrolytic reaction of the equi-mass binary mixtures can be considered a first-order reaction.

Similarly, as presented in Table 8, the third-order reaction (F3) mechanism is the most suitable reaction mechanism for the pyrolysis of the equi-mass ternary mixtures of PS, PP, and LDPE. 

Generally, LDPE has a higher activation energy than that of pure PS and PP. The addition of the third polymer (LDPE) resulted in a slower reaction rate (third-order reaction with a higher activation energy) and a higher energy needed for the reaction (higher activation energy). The change in reaction mechanism could also be attributed to the interaction between the mixture components, resulting in a synergistic effect [15].

Practically, the pyrolysis of the binary mixtures of PS and PP needs a lower amount of energy for the reaction to take place (lower *E_a_* value) and has a faster reaction rate (first-order reaction with lower activation energy) when compared to the pyrolysis of ternary mixtures of PS, PP, and LDPE. Therefore, pyrolysis of binary mixtures is preferable. 

### 3.4. Pre-Exponential Factor Determination

After the determination of the most suitable reaction mechanism, the pre-exponential factor was obtained using the Friedman, FWO, KAS, and Starink isoconversional models. Table 9 and Table 10 illustrate the values of *ln(A)* binary and ternary polymer mixtures.

To ascertain the suitability of the reaction mechanism, a linear relationship between *ln(A)* and *E_a_* was established [21]. Figure 6 proves the perfect linear relationship between *ln(A)* and *E_a_* obtained using all isoconversional models (R^2^ = 0.9959). This finding confirms the appropriateness of the obtained reaction mechanisms. The triplet kinetic parameters for the pyrolytic reactions of the binary (PS and PP) and ternary (PS, PP, LDPE) mixtures are summarized in Table 11.

To continue this work, an artificial neural network model can be developed to predict the TGA data [22]. In addition, sensitivity analysis can be performed to explore the relationship between the input and output parameters.

## 4. Conclusions

The pyrolytic kinetics and mechanism of polystyrene (PS), polypropylene (PP), and low-density polyethylene (LDPE) mixtures, which represent almost half of the municipal plastic waste (MPW), have been investigated. This research will contribute significantly to the proper treatment of the huge waste quantity that threatens our environment. Specifically, this work aims to develop a kinetic model that can fully describe the thermal decomposition of plastic mixtures.

In this work, pyrolysis of different plastic mixtures, consisting of PS, PP, and LDPE, was performed using a thermogravimetric analyzer (TGA) at three different heating rates (5, 20, and 40 K/min). Four isoconversional models, namely Friedman, Flynn–Wall–Qzawa (FWO), Kissinger–Akahira–Sunose (KAS), and Starink, have been used to obtain the kinetic parameters of the pyrolysis of different plastic mixtures with different compositions. For the equi-mass binary mixtures of PP and PS, the average values of the activation energies were 181, 144 ± 2 kJ/mol obtained by the Freidman and integral (FWO, KAS, and Starink) models, respectively. However, higher values were obtained for the equi-mass ternary plastic mixtures of PP, PS, and LDPE (Freidman: 255 kJ/mol, FWO: 222 kJ/mol, KAS: 223 kJ/mol, and Starink: 222 kJ/mol). Then, the best suitable reaction mechanism was obtained using the Coats–Redfern model. The results confirm that the most controlling reaction mechanisms obey the first-order and third-order reactions for the pyrolysis of the equi-mass binary (PS and PP) and equi-mass ternary (PS, PP, and LDPE) mixtures, respectively. Finally, the values of the pre-exponential factor were obtained using the four isoconversional models, and the linear relationship between *ln A*, and the obtained activation energy was confirmed. The results reveal that the pyrolysis of binary mixtures is preferable, with lower energy consumption and a faster reaction rate. 

## Figures and Tables

**Figure 1 polymers-13-03413-f001:**
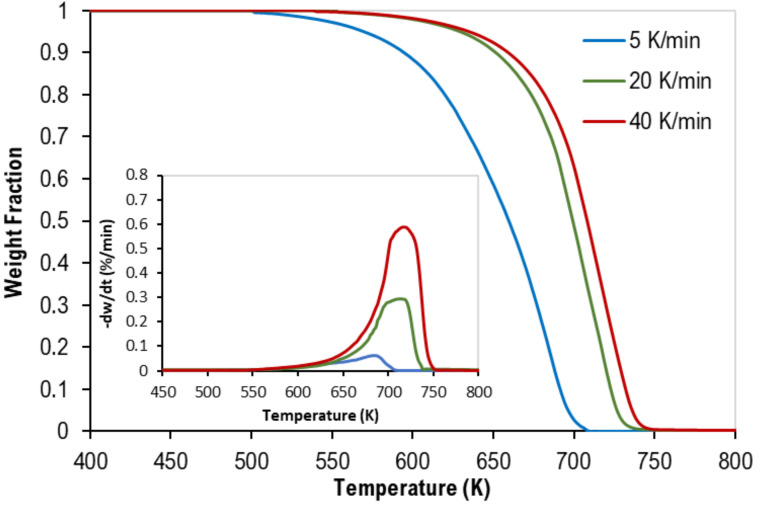
Thermogravimetric (TG) curves of equi-mass binary mixtures of PP and PS (Tests 1–3). Inset: corresponding derivative thermogravimetric (DTG) curves.

**Figure 2 polymers-13-03413-f002:**
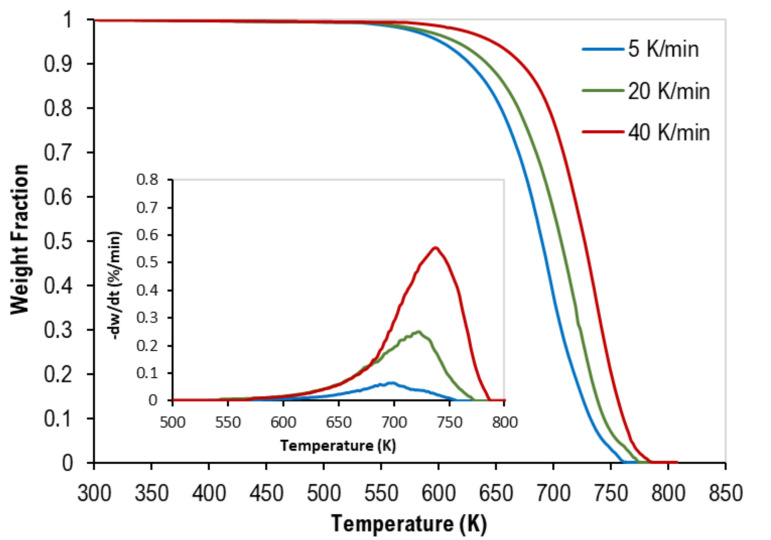
Thermogravimetric (TG) curves of equi-mass ternary mixtures of PP, PS, and LDPE (Tests 4–6). Inset: corresponding derivative thermogravimetric (DTG) curves.

**Figure 3 polymers-13-03413-f003:**
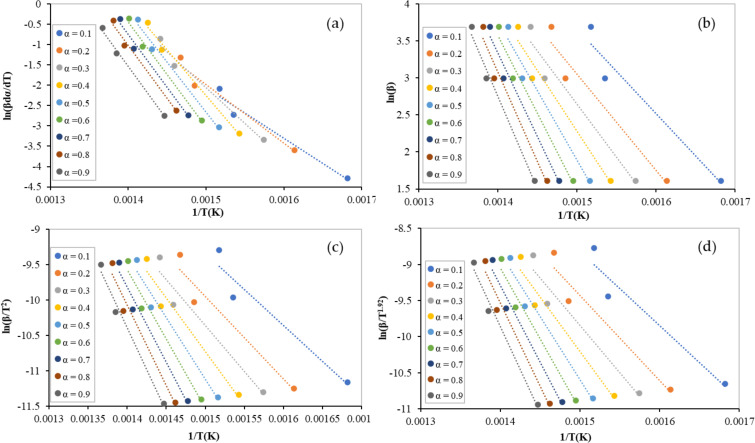
Linear regression lines of binary equi-mass PP and PS mixtures at different conversions by: (**a**) Freidman; (**b**) FWO; (**c**) KAS; and (**d**) Starink.

**Figure 4 polymers-13-03413-f004:**
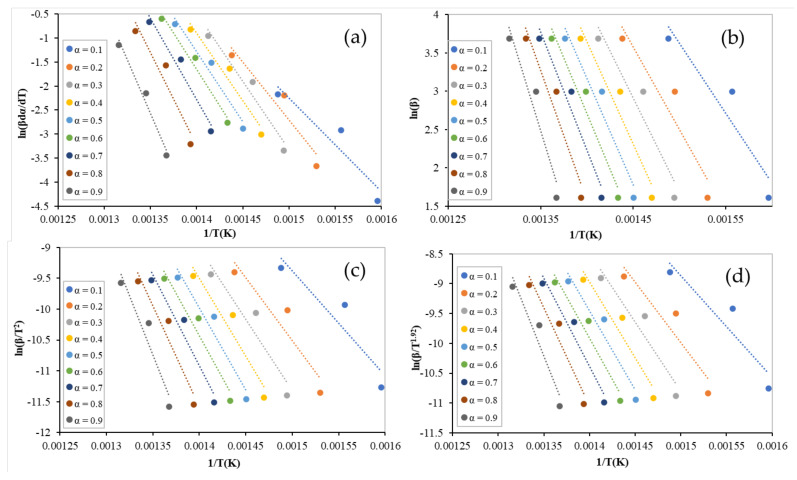
Linear regression lines of ternary equi-mass PP, PS, and LDPE mixtures at different conversions by: (**a**) Freidman; (**b**) FWO; (**c**) KAS; and (**d**) Starink.

**Figure 5 polymers-13-03413-f005:**
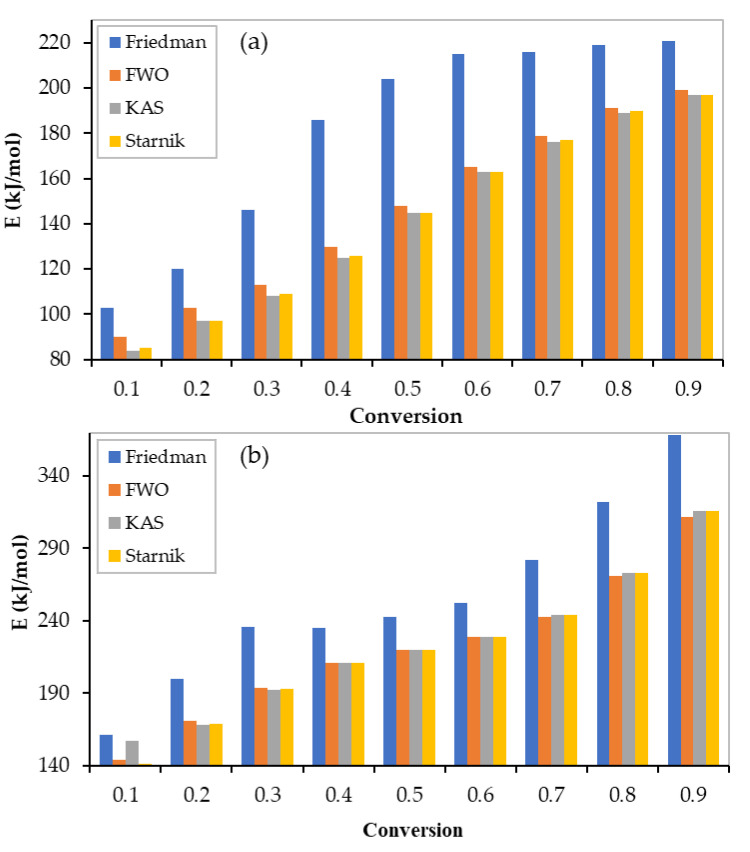
Activation energies obtained by different model-free models: (**a**) binary mixtures; (**b**) ternary mixtures.

**Figure 6 polymers-13-03413-f006:**
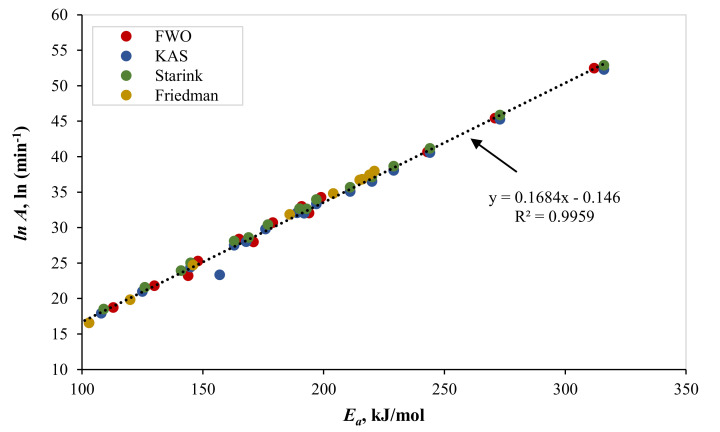
Linear relationship between ln *A* and *Ea*.

**Table 1 polymers-13-03413-t001:** Proximate and ultimate analysis of the used plastics.

Plastic	Proximate Analysis, wt%			Ultimate Analysis, wt%			
	Moisture	Volatile	Ash	C	H	N	S
PP	0.08	99.63	0.29	85.00	14.73	0.04	0.23
PS	0.24	99.59	0.17	90.47	9.43	0.00	0.08
LDPE	0.20	99.65	0.15	83.00	16.75	0.00	0.25

**Table 2 polymers-13-03413-t002:** Experimental matrix.

Test No.	Heating Rate (K/min)	Weight %		
		PP	PS	LDPE
1	5	50	50	0
2	20	50	50	0
3	40	50	50	0
4	5	33.3	33.3	33.3
5	20	33.3	33.3	33.3
6	40	33.3	33.3	33.3

**Table 3 polymers-13-03413-t003:** Solid-state thermal reaction mechanisms.

Reaction Mechanism	$f\left(\alpha \right)$	$g\left(\alpha \right)$
** *Reaction-Order Models* **		
First order (F1)	1-α	$-\ln\left(1-\alpha \right)$
Second order (F2)	${\left(1-\alpha \right)}^{2}$	${\left(1-\alpha \right)}^{-1}-1$
Third order (F3)	${\left(1-\alpha \right)}^{3}$	$[{\left(1-\alpha \right)}^{-1}-1]/2$
** *Diffusion Models* **		
One-dimensional (D1)	$1/2\alpha^{-1}$	$\alpha^{2}$
Two-dimensional (D2)	${\left[-ln\left(1-\alpha \right)\right]}^{-1}$	$\left(1-\alpha \right)\ln\left(1-\alpha \right)+\alpha$
Three-dimensional (D3)	$3/2{\left[1-{\left(1-\alpha \right)}^{1/3}\right]}^{-1}$	${\left[1-{\left(1-\alpha \right)}^{1/3}\right]}^{2}$
** *Nucleation Models* **		
Two-dimensional nucleation (A2)	$2\left(1-\alpha \right){\left[-\ln\left(1-\alpha \right)\right]}^{1/2}$	${\left[-\ln\left(1-\alpha \right)\right]}^{1/2}$
Three-dimensional nucleation (A3)	$3\left(1-\alpha \right){\left[-\ln\left(1-\alpha \right)\right]}^{1/3}$	${\left[-\ln\left(1-\alpha \right)\right]}^{1/3}$
Four-dimensional nucleation (A4)	$4\left(1-\alpha \right){\left[-\ln\left(1-\alpha \right)\right]}^{1/4}$	${\left[-\ln\left(1-\alpha \right)\right]}^{1/4}$
** *Geometrical Contraction Models* **		
Prout–Tompkins (R1)	1	α
Contracting cylinder (R2)	$2{\left(1-\alpha \right)}^{1/2}$	$1-{\left(1-\alpha \right)}^{1/2}$
Contracting sphere (R3)	$3{\left(1-\alpha \right)}^{1/3}$	$1-{\left(1-\alpha \right)}^{1/3}$
** *Power Law Models* **		
Power law (P2)	$2\alpha^{1/2}$	$\alpha^{1/2}$
Power law (P3)	$3\alpha^{2/3}$	$\alpha^{1/3}$
Power law (P4)	$4\alpha^{3/4}$	$\alpha^{1/4}$

**Table 4 polymers-13-03413-t004:** Onset, peak, and final decomposition temperatures.

Test No.	Heating Rate (K/min)	Weight %			Onset Temp. (K)	Peak Temp. (K)	Final Temp. (K)
		PP	PS	LDPE			
1	5	50	50	0	500	686	708
2	20	50	50	0	550	720	734
3	40	50	50	0	560	730	750
4	5	33.3	33.3	33.3	540	690	760
5	20	33.3	33.3	33.3	610	720	775
6	40	33.3	33.3	33.3	630	735	785

**Table 5 polymers-13-03413-t005:** Activation energy of the binary mixtures of PP and PS obtained by different model-free models.

Conversion	Differential Method		Integral Methods							
	Friedman		FWO		KAS		Starink		Average Values	
	$E_{a}\;(\mathrm{kJ}/\mathrm{mol})$	R^2^	$E_{a}\;(\mathrm{kJ}/\mathrm{mol})$	R^2^	$E_{a}\;(\mathrm{kJ}/\mathrm{mol})$	R^2^	$E_{a}\;(\mathrm{kJ}/\mathrm{mol})$	R^2^	$E_{a}\;(\mathrm{kJ}/\mathrm{mol})$	R^2^
0.1	103	0.9663	90	0.9458	84	0.9329	85	0.9335	86	0.9374
0.2	120	0.9668	103	0.9525	97	0.9423	97	0.9428	99	0.9459
0.3	146	0.9803	113	0.9562	108	0.9476	109	0.948	110	0.9506
0.4	186	0.9911	130	0.9641	125	0.9579	126	0.9582	127	0.9601
0.5	204	0.9887	148	0.9709	145	0.9665	145	0.9667	146	0.9680
0.6	215	0.9915	165	0.9756	163	0.9723	163	0.9724	164	0.9734
0.7	216	0.9852	179	0.9783	176	0.9755	177	0.9756	177	0.9765
0.8	219	0.9888	191	0.9724	189	0.9691	190	0.9693	190	0.9703
0.9	221	0.995	199	0.9867	197	0.9852	197	0.9852	198	0.9857
**Average**	**181**	**0.9837**	**146**	**0.9669**	**142**	**0.961**	**143**	**0.9613**	**144**	**0.9631**

**Table 6 polymers-13-03413-t006:** Activation energy of the ternary mixtures of PP, PS, and LDPE obtained by different model-free models.

Conversion	Differential Method		Integral Methods							
	Friedman		FWO		KAS		Starink		Average Values	
	E (kJ/mol)	R^2^	$E_{a}\;(\mathrm{kJ}/\mathrm{mol})$	R^2^	$E_{a}\;(\mathrm{kJ}/\mathrm{mol})$	R^2^	$E_{a}\;(\mathrm{kJ}/\mathrm{mol})$	R^2^	$E_{a}\;(\mathrm{kJ}/\mathrm{mol})$	R^2^
0.1	161	0.8895	144	0.8859	157	0.8696	141	0.8703	147	0.8753
0.2	200	0.9164	171	0.898	168	0.8852	169	0.8858	169	0.8897
0.3	236	0.9549	194	0.9164	192	0.9069	193	0.9073	193	0.9102
0.4	235	0.9561	211	0.9363	211	0.9294	211	0.9297	211	0.9318
0.5	243	0.9645	220	0.948	220	0.9424	220	0.9427	220	0.9444
0.6	252	0.9731	229	0.9562	229	0.9516	229	0.9518	229	0.9532
0.7	282	0.9608	243	0.9563	244	0.952	244	0.9522	244	0.9535
0.8	322	0.9232	271	0.9395	273	0.9342	273	0.9344	272	0.9360
0.9	368	0.9749	312	0.9265	316	0.921	316	0.9212	315	0.9229
**Average**	**255**	**0.9459**	**222**	**0.9292**	**223**	**0.9213**	**222**	**0.9217**	**222**	**0.9241**

**Table 7 polymers-13-03413-t007:** Kinetic parameters of the pyrolysis of plastic mixtures by Coats–Redfern model (tests 1–3).

Reaction Mechanism	5 K/min		20 K/min		40 K/min		Average Values	
	$E_{a}\;(\mathrm{kJ}/\mathrm{mol})$	*R^2^*	$E_{a}\;(\mathrm{kJ}/\mathrm{mol})$	*R^2^*	$E_{a}\;(\mathrm{kJ}/\mathrm{mol})$	*R^2^*	$E_{a}\;(\mathrm{kJ}/\mathrm{mol})$	R^2^
F1	87	0.9998	182	0.9992	188	0.9997	**152**	**0.9996**
F2	110	0.9981	259	0.9933	271	0.9945	**213**	**0.9953**
F3	137	0.9941	352	0.9837	372	0.9863	**287**	**0.9880**
D1	144	0.9976	258	0.995	261	0.9958	**221**	**0.9961**
D2	156	0.9989	292	0.9977	297	0.9983	**248**	**0.9983**
D3	170	0.9996	333	0.9992	341	0.9997	**281**	**0.9995**
A2	38	0.9998	85	0.9991	88	0.9996	**70**	**0.9995**
A3	22	0.9997	53	0.999	55	0.9996	**43**	**0.9994**
A4	14	0.9995	37	0.9989	38	0.9996	**30**	**0.9993**
R1	67	0.9972	123	0.9945	125	0.9954	**105**	**0.9957**
R2	76	0.9992	151	0.9986	154	0.9992	**127**	**0.9990**
R3	80	0.9996	161	0.9992	165	0.9997	**135**	**0.9995**
P2	28	0.9957	56	0.9932	56	0.9943	**47**	**0.9944**
P3	15	0.9931	33	0.9914	34	0.9928	**27**	**0.9924**
P4	9	0.9877	22	0.9888	22	0.9906	**18**	**0.9890**

**Table 8 polymers-13-03413-t008:** Kinetic parameters of the pyrolysis of plastic mixtures by Coats–Redfern model (tests 4–6).

Reaction Mechanism	5 K/min		20 K/min		40 K/min		Average Values	
	$E_{a}\;(\mathrm{kJ}/\mathrm{mol})$	*R^2^*	$E_{a}\;(\mathrm{kJ}/\mathrm{mol})$	*R^2^*	$E_{a}\;(\mathrm{kJ}/\mathrm{mol})$	*R^2^*	$E_{a}\;(\mathrm{kJ}/\mathrm{mol})$	R^2^
F1	120	0.9996	114	0.9992	143	0.9999	**126**	**0.9996**
F2	176	0.9962	152	0.9962	181	0.9993	**170**	**0.9972**
F3	244	0.9903	196	0.9924	224	0.9971	**221**	**0.9933**
D1	165	0.9955	176	0.9998	233	0.998	**191**	**0.9978**
D2	190	0.9979	195	1	253	0.9989	**213**	**0.9989**
D3	220	0.9994	217	0.9998	275	0.9996	**237**	**0.9996**
A2	54	0.9996	51	0.9991	66	0.9999	**57**	**0.9995**
A3	32	0.9995	30	0.9989	40	0.9998	**34**	**0.9994**
A4	21	0.9993	20	0.9986	27	0.9998	**23**	**0.9992**
R1	77	0.9948	82	0.9997	111	0.9977	**90**	**0.9974**
R2	97	0.9987	97	0.9999	126	0.9993	**107**	**0.9993**
R3	104	0.9993	103	0.9998	132	0.9996	**113**	**0.9996**
P2	33	0.9926	35	0.9996	49	0.9971	**39**	**0.9964**
P3	18	0.9889	20	0.9994	29	0.9961	**22**	**0.9948**
P4	11	0.9817	12	0.999	19	0.9947	**14**	**0.9918**

**Table 9 polymers-13-03413-t009:** Pre-exponential factor of the co-pyrolysis of PS and PP (tests 1–3).

Conversion	*ln (A)* (ln min^−1^)			
	Friedman	FWO	KAS	Starink
0.1	16.58	14.09	13.41	14.00
0.2	19.81	16.58	15.82	16.41
0.3	24.70	18.72	17.92	18.52
0.4	31.85	21.81	20.97	21.57
0.5	34.81	25.28	24.42	25.02
0.6	36.71	28.39	27.50	28.10
0.7	36.79	30.71	29.80	30.40
0.8	37.44	32.98	32.06	32.66
0.9	37.96	34.28	33.34	33.94
**Average**	**30.74**	**24.76**	**23.92**	**24.52**

**Table 10 polymers-13-03413-t010:** Pre-exponential factor of the co-pyrolysis of PS, PP, and LDPE (tests 4-6).

Conversion	*ln (A)* (ln min^−1^)			
	Friedman	FWO	KAS	Starink
0.1	27.19	23.22	23.32	23.92
0.2	34.13	27.96	27.99	28.59
0.3	40.38	32.04	32.03	32.63
0.4	40.21	35.12	35.07	35.68
0.5	41.66	36.57	36.50	37.10
0.6	43.50	38.15	38.06	38.66
0.7	48.80	40.67	40.55	41.16
0.8	55.84	45.41	45.26	45.87
0.9	64.05	52.45	52.28	52.89
**Average**	**43.97**	**36.84**	**36.78**	**37.39**

**Table 11 polymers-13-03413-t011:** Kinetic parameters of the pyrolysis of mixed polymers.

Mixture	Binary		Ternary		
Polymers	PS	PP	PS	PP	LDPE
Composition (wt %)	50	50	33.3	33.3	33.3
Reaction Mechanism	F1		F3		
$E_{a}$ **(kJ/mol)**					
Differential Model (Friedman)	181		255		
Integral Models (FWO, KAS, and Starink)	144 ± 2 *		222.3 ± 0.6 *		
***ln A* (ln min** **^−1^)**					
Differential Model (Friedman)	30.74		43.97		
Integral Models (FWO, KAS, and Starink)	24.4 ± 0.4 *		37 ± 0.3 *		

* mean value ± standard deviation.

## Data Availability

The data presented in this study are available in the article.

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
