# Peer review of "Thermal Behavior of Mixed Plastics at Different Heating Rates: I. Pyrolysis Kinetics"

_polymers, 2021, doi:10.3390/polym13193413_

Round 1
Reviewer 1 Report
This work aims to develop kinetic model describing thermal decomposition of plastic mixtures. The investigations are connected with plastic waste management and would interesting in this field. The paper could be reconsidered after the revision:
*The 2 sentences are in Abstract ?
Then, the best suitable reaction mechanisms were obtained using Coats-Redfern model. Then, the best suitable reaction mechanism was obtained using Coats-Redfern model.
*In the municipal plastic waste there are different amounts of different plastics and also other materials/ additives, also different amounts of the mentioned polystyrene (PS), polypropylene (PP) and low-density polyethylene (LDPE). The different composition and the additives could strongly influence the investigations described in the manuscript and the obtained results of investigation would depend on composition of the waste and would be un -repeatable. Usually all polymeric products have additives, which would influence the processes investigated here. What would be opinion of the authors about this ? Would the theoretical models suitable for real plastic waste ?
*Introduction of the paper is rather long and described what is done with other plastics or plastic waste in the field. The authors should concentrate the introduction for researcher in field of plastics comprising of PP, PS, and LDPE.
*This works aims to develop a kinetic model that can fully describe the thermal behavior of the pyrolysis of mixed plastics comprising of PP, PS, and LDPE at different heating rates. In addition, the triple kinetic parameters (activation energy, pre-exponential factor, and most suitable controlling mechanism/s) have been determined. Author should explain a practical value of the investigations in introduction or conclusions. Is the kinetic model and parameters useful for practical works with plastic waste ?
* The investigations were done with pellets of PP, PS, and LDPE, supplied by Ipoh SY Recycle Plastic Co. Would it be possible to investigate under the conditions real plastic waste with additives as they are in polymeric products?
* Why the homopolymers: polystyrene (PS), polypropylene (PP) and low-density polyethylene (LDPE) are not tested here for comparison together with the polymer mixtures ?
Author Response
Dear Respected Reviewer,
Thanks for your valuable comments.
Please find attached the authors' response to your comments.
Best Regards

Reviewer 2 Report
The paper presents a study of the thermal decomposition of plastics via thermogravimetric analysis. Several polymers and mixtures thereof commonly found in municipal waste are studied. A range of models are fitted to the data to determine the reaction kinetics and Arrhenius parameters. Overall the paper has been well-written and I recommend it for publication in Polymers. Some clarifications about the kinetic modelling would further enhance the paper:
- It would be helpful to give some further information about how the kinetic parameters are extracted by fitting the Coates Redfern model to the data. Moreover, how the solid-state thermal reaction mechanisms from Table 4 have been investigated in conjunction with the Coates Redfern model? The models in Table 4 are only briefly referred to at Line 162.
- The linear regression plots at Figures 3 and 4 only have 3 data points for each conversion, and thus some of the lines don’t look to be a particularly good fit to the straight line. Given the large amount of data generated from TGA, is it possible to include extra points to verify the fit of some of the lines?
- For the reaction mechanism determination on page 9, it appears that the kinetic model that best matches the activation energy of the isoconversional models is selected as the best fitting model. Perhaps some further discussion of why this approach is valid would be beneficial here?
- The discussion is limited to saying which model fitted best statistically, e.g. first order for the binary and third order for the ternary mixtures. Perhaps the discussion could be extended to try to offer some explanation as to why this occurs, for example, synergistic effects or more complex reactions that occur as more polymers are added to the mixture.
Author Response

(The authors gave the same response as above.)

Round 2
Reviewer 1 Report
I recommend the revised version for publication if other reviewers and editor think that the paper is suitable for “Polymers” ?